# There Are No Shortcuts to Anywhere Worth Going: Identifying Shortcuts in Deep Learning Models for Medical Image Analysis

**Christopher Boland**[1,2]          CHRISTOPHER.BOLAND@MRE.MEDICAL.CANON

**Keith A. Goatman**[1]          KEITH.GOATMAN@MRE.MEDICAL.CANON

**Sotirios A. Tsaftaris**[2]          S.TSAFTARIS@ED.AC.UK

**Sonia Dahdouh**[1]          SONIA.DAHDOUH@MRE.MEDICAL.CANON

[1] *Canon Medical Research Europe, Edinburgh, EH6 5NP, UK*

[2] *School of Engineering, The University of Edinburgh, Edinburgh, EH9 3FG, UK*

**Editors:** Accepted for publication at MIDL 2024

## Abstract

Many studies have reported human-level accuracy (or better) for AI-powered algorithms performing a specific clinical task, such as detecting pathology. However, these results often fail to generalize to other scanners or populations. Several mechanisms have been identified that confound generalization. One such is shortcut learning, where a network erroneously learns to depend on a fragile spurious feature, such as a text label added to the image, rather than scrutinizing the genuinely useful regions of the image. In this way, systems can exhibit misleadingly high test-set results while the labels are present but fail badly elsewhere where the relationship between the label and the spurious feature breaks down. In this paper, we investigate whether it is possible to detect shortcut learning and locate where the shortcut is happening in a neural network. We propose a novel methodology utilizing the sample difficulty metric Prediction Depth (PD) and KL divergence to identify specific layers of a neural network where the learned features of a shortcut manifest. We demonstrate that our approach can isolate these layers for several shortcuts, model architectures, and datasets. Using this, we show a correlation between a shortcut's visual complexity, the depth of its feature manifestation within the model, and it's impact to model performance. Finally, we highlight the nuanced relationship between learning rate and shortcut learning.

**Keywords:** shortcut learning, prediction depth, spurious correlations, model robustness.

## 1. Introduction

Shortcut learning is the phenomenon where deep neural networks rely on superficial or irrelevant data features, termed 'shortcuts' (Ahmed et al., 2022). Such features are easy to learn but do not generalize beyond the training data and performance degrades post-deployment (Cohen et al., 2020; Pooch et al., 2020; Zech et al., 2018; Beery et al., 2018). Models can rely on these spurious features even if they are less predictive than those that are clinically relevant (Shah et al., 2020). In many cases, the shortcut features may not be recorded in the dataset, making it challenging to identify when a model relies on them (Oakden-Rayner et al., 2020). This poses a significant challenge to clinical deployment of medical image analysis systems.

Unfortunately, shortcuts are common in medical image datasets. For example, Nauta et al. (2021) found that models trained using a popular skin lesion dataset exploit the

color calibration charts to make predictions, since they are only present in the malignant cases. Similarly, patients suffering a pneumothorax often have chest drains installed that are visible in chest x-rays; classification models can use these to make predictions (Jiménez-Sánchez et al., 2023). Even subtle features, such as small differences in the image acquisition process or patient age, can introduce shortcuts (Ahmed et al., 2022; Brown et al., 2023). Our understanding of shortcut learning remains limited. Recent works propose targeting features learned in the earliest layers to mitigate shortcuts (Murali et al., 2023; Dagaev et al., 2023). This overlooks shortcut features that may manifest deeper within the model.

This paper presents three main contributions. Firstly, we propose and demonstrate the efficacy of a novel methodology to localize learned shortcuts to specific model layers. We utilize a metric called Prediction Depth (PD) (Baldock et al., 2021) and KL Divergence to identify changes in the model's behavior after adding a shortcut to the training data. We show that our approach can consistently and effectively localize learned shortcuts to specific model layers across various architectures, shortcut types, and natural or medical image tasks. Secondly, we evaluate the relationship between a shortcut's complexity, how deep in the model it manifests, and its potential for harm. Finally, we investigate the impact of training condition hyperparameters, such as learning rate, on shortcut learning.

## 2. Preliminaries

### 2.1. Prediction depth

Prediction depth (PD) quantifies example difficulty by the number of layers a model needs to finalize its prediction. k-NN classification probes provide a per-layer prediction for each sample. PD is defined as the layer after which all subsequent probes make the same prediction. More challenging inputs require disambiguation of more complex features and would be expected to have deeper PD (Baldock et al., 2021). Murali et al. (2023) connect PD to shortcut learning and the simplicity bias of deep neural networks (Shah et al., 2020), demonstrating that shortcuts are harmful when they are simpler than the relevant features.

### 2.2. Kullback-Leibler (KL) divergence

KL divergence is a measure of dissimilarity between two probability distributions. This is expressed in Equation (1) where $X$ is a collection of events; $P(x)$ is the probability of event $x$ occurring in distribution $P$; and $Q(x)$ is the probability of $x$ occurring in distribution $Q$.

$$D_{KL}(P(x)||Q(x)) = \sum_{x \in X} P(x) \log_2 \frac{P(x)}{Q(x)} \tag{1}$$

## 3. Method

### 3.1. Shortcut localization

We localize the network layer specific to learned shortcuts based on KL divergence and PD. To illustrate this, we train two binary classification models. One is trained on a dataset with no known shortcuts and the other on a dataset with a known shortcut which is perfectly correlated with one class. Both models are evaluated on the same test set, where the

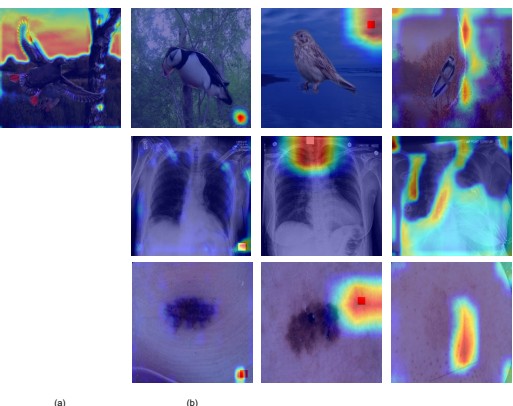

Figure 1: Heatmaps for samples identified using our shortcut localization method. Samples are classified by a ResNet-34, applied to three distinct datasets. Each sample contains a shortcut that the model uses to make its prediction. We test across several shortcuts: (a) background (waterbirds only); (b) red square (constant location); (c) red square (random location); and (d) complex object.

shortcut is balanced between both classes. We fit k-NN probes, with k=29 to avoid ties, to each convolutional layer. This provides a per-layer classification for each sample, from which we obtain a PD distribution over the entire set.

To isolate specific layers where the shortcut manifests, we calculate the KL divergence between the two PD distributions (Equation (1)). Each "event", $x$, is a possible prediction depth, and $P$ and $Q$ are the distributions of the shortcut and clean models, respectively. Rather than summing the divergence of every PD we consider each individually. The asymmetric nature of KL divergence means events that are more likely in $P$ than $Q$ lend more weight to the divergence than those more likely in $Q$ than $P$. This emphasizes the layers most influenced by the shortcut. For this reason, it was chosen over other metrics, such as the square difference. We isolate the layers that exhibit the most significant divergence (the 95[th] percentile). GradCAM heatmaps (Selvaraju et al., 2017), provide an intuition of the shortcut's influence at these layers (Figure 1). While the link between saliency techniques and model outputs is debated, and they are vulnerable to adversarial attacks (Zhang et al., 2022), GradCAM has been shown to be valuable for interpreting model decisions due to its sensitivity to model parameters and training data labels (Adebayo et al., 2018).

### 3.2. Introducing synthetic shortcuts

To probe how different types of shortcut influence models, we generate augmented datasets introducing three types of synthetic shortcuts (Figure 2):

1. **Background:** Leveraging image background as a spurious signal (only Waterbirds).

2. **Simple object:** Introducing small, discrete objects to the image. We test two variants:

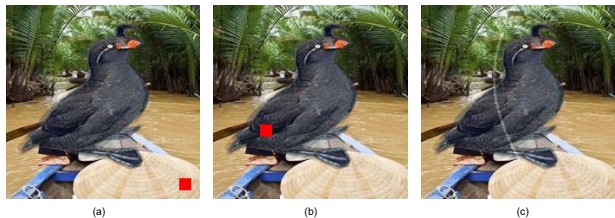

Figure 2: Image from the Waterbirds dataset augmented with synthetic shortcuts: (a) red square (constant location); (b) red square (random location); (c) complex object.

- **Constant location:** The object appears in a fixed spot.
- **Random location:** The object location varies among images.

3. **Complex object:** Inspired by real-world objects that commonly trigger shortcuts (e.g. chest drains in pneumothorax-positive X-ray images) we introduce a vertical, semi-opaque curved line, centered in the image. To simulate real-world stochasticity we introduce random rotations and allow for variation in its location.

These shortcuts vary in visual complexity: background shortcuts offer a ubiquitous, low-level texture cue; more complex shortcuts introduce higher-level features that require more effort to disambiguate.

### 3.3. Datasets and models

We test our approach to shortcut localization on three datasets: Waterbirds, a subset of CheXpert, and ISIC 2017 (Sagawa et al., 2020; Jiménez-Sánchez et al., 2023; Codella et al., 2018). Waterbirds introduces a background shortcut and is commonly used for evaluating spurious correlations and out of distribution (OOD) performance. Both CheXpert and ISIC are popular public medical image datasets that have been studied in the context of bias and shortcut learning (Brown et al., 2023; Nauta et al., 2021). For each dataset, we consider a binary classification task, apply class balancing by undersampling, and use the train/test/validation splits provided by the authors. For CheXpert, we predict the presence of a pneumothorax and with ISIC we classify lesions as benign or malignant. All images are resized to $224 \times 224$ and random rotation (0 to 10° around the image center) is applied.

Four popular model architectures are used: ResNet-34 (He et al., 2016), DenseNet-121 (Huang et al., 2017), InceptionNet (Szegedy et al., 2016), and VGG16 (Simonyan and Zisserman, 2014). SGD and AdamW optimizers are tested (Loshchilov and Hutter, 2017) with three learning rates: 0.01, 0.001, and 0.0001 with early stopping on the validation loss.

### 4. Experiments

To validate our approach to shortcut localization, and enhance our understanding of shortcut learning, we conduct three experiments. Firstly, we show that adding a shortcut to training data significantly alters the mean PD of the test set. Secondly, we demonstrate

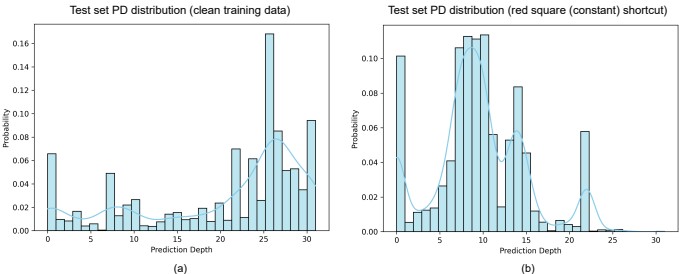

Figure 3: Test-set PD distributions of a ResNet-34 trained on: (a) clean dataset; (b) dataset featuring "red square (constant)" shortcut. In the test set the shortcut is evenly distributed between classes. Model (b) has a distinct peak around the $7^{\text{th}}$-$10^{\text{th}}$ layers and almost no samples classified in the later layers, suggesting it uses simpler features to make predictions.

Table 1: P-value of a Welch's t-test measuring the difference between the mean test set PD of the clean model and each of the shortcut models.

| Shortcut | ResNet-34 | DenseNet-121 | VGG-16 | InceptionNet |
|---|---|---|---|---|
| Background | $6.021 \times 10^{-20}$ | $4.869 \times 10^{-13}$ | $2.600 \times 10^{-8}$ | $1.089 \times 10^{-11}$ |
| Red square (constant) | 0.0 | 0.0 | 0.0 | $6.497 \times 10^{-54}$ |
| Red square (random) | $2.242 \times 10^{-11}$ | $2.791 \times 10^{-05}$ | 0.0 | 0.0 |
| Complex object | $7.807 \times 10^{-144}$ | $1.532 \times 10^{-123}$ | $2.005 \times 10^{-121}$ | 0.103 |

that the divergence of the shortcut model's PD distribution from the clean model's can identify the layers where the shortcut most influences model behavior. Thirdly, we explore the relationship between the visual complexity of shortcuts, the depth at which they are learned, and the harm they cause (how much they reduce performance and hurt robustness).

### 4.1. Shortcuts reduce prediction depth

We hypothesize that including a shortcut in the training data causes a change to the PD distribution. To validate this, we use the Waterbirds dataset (Sagawa et al., 2020) and focus on the binary classification task Waterbird/Landbird. We generate a new, class-balanced, "clean" dataset without the background shortcut. Following Section 3.2, we augment this dataset with synthetic shortcuts (Figure 2), producing three new train, validation, and test sets. Models are trained with an SGD optimizer at a learning rate of 0.01.

We compare the test set PD distribution of each shortcut model with the clean benchmark (Figure 3). When making this comparison, the clean model is evaluated on the test set featuring the shortcut. Using Welch's t-test (that does not assume equal distribution variances), we demonstrate that there is a statistically significant change in the mean PD of the test set when a shortcut is introduced to the training data ($p \ll 0.05$) (see Table 1).

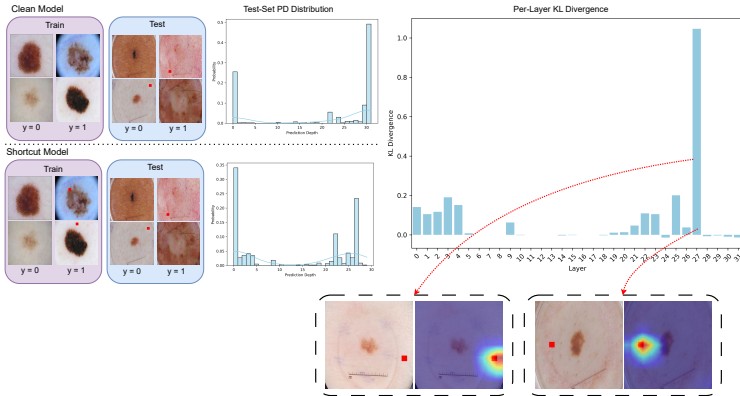

Figure 4: Proposed method for shortcut localization. Identical models are trained on the same dataset, but for one the images are augmented with a synthetic shortcut. In the test set the shortcut is balanced between classes. KL divergence analysis of the PD distribution of the shortcut model from the clean model highlights the layers with the most significant change from the clean model. Inspection of samples classified at this layer shows that the shortcut is used.

### 4.2. Shortcuts can be localized to specific layers

Knowing that shortcuts influence the PD, we demonstrate that a learned shortcut can be localized to particular layers using KL divergence analysis. We test this on the Waterbirds dataset and two medical image datasets (CheXpert and ISIC). Similar to Section 4.1, we create three shortcut datasets each for CheXpert and ISIC, using the original datasets as "clean" controls. For each model trained on the shortcut datasets, we isolate the layers that exhibit the most significant divergence from the PD of the clean model (Figure 4).

The GradCAM heatmaps showcased in Figure 1 illustrate that shortcuts impact predictions for samples identified in layers with the greatest divergence. This is consistent across all datasets and shortcut types. Figure 4 and Figure 5 provide a more detailed illustration of our approach applied to ISIC and CheXpert examples. We consistently find that the layers with the highest divergence in the model corresponded to samples whose predictions were influenced by the presence or absence of shortcuts. By utilizing KL divergence, we pinpoint the layers where the shortcuts were most pronounced, even in cases where a simple inspection of the PD distribution might not have been enough. For example, in Figure 4, the shortcut model's largest peak in the PD distribution is in layer 0. However, by using KL divergence, we were able to identify that the layer most impacted by the shortcut was actually layer 27. Further inspection of samples with a PD of 27 revealed that the model was indeed leveraging the shortcut in order to make its predictions.

### 4.3. Shortcuts are not all created equal

We consider the relationship between a shortcut's visual complexity, the depth at which it is learned, and the harm it causes. We anticipate that more complex shortcuts will

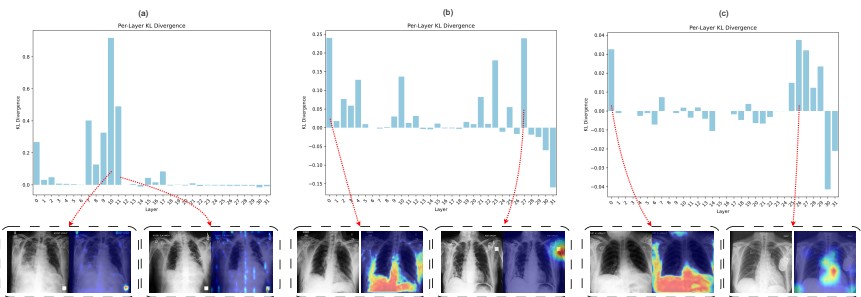

Figure 5: Comparison of depth of shortcut manifestations: (a) red square (constant); (b) red square (random); (c) complex object. We highlight that layers with high divergence indicate the influence of the shortcut. In (b) and (c), high divergence in early layers correlate with samples where the shortcut is absent. More complex shortcuts manifest in the later layers.

require more effort to disambiguate and will, therefore, skew the test set PD distribution deeper. To evaluate this, we take our shortcut models from Section 4.2 and compare their performance, divergence from the benchmark, and mean PD. Furthermore, we explore the impact of learning rate on the potential harm caused by shortcuts.

Prior work suggests that shortcuts manifest in the earliest layers of a model, and can be mitigated by targeting these layers (Murali et al., 2023; Tiwari et al., 2024). In Figures 4 and 5 we demonstrate that shortcuts are not constrained to the early layers. More visually complex shortcuts tend to manifest deep in the model. We see that the red square (random) and complex object shortcuts have a significant influence on the 27[th] and 26[th] layers respectively. Inspection of samples classified at these layers shows that the shortcut is used for their prediction. We also see a peak in the earliest layer, where we see more samples that do not contain the shortcut are classified. Mitigation that focuses purely on the earliest layers may fail to address more visually complex shortcuts.

Inspection of test-set PD distribution of models trained on our datasets highlights clear differences in how shortcuts manifest within the same model. Our findings, illustrated in Table 2, indicate that as shortcut complexity increases, performance cost decreases. The simplest square shortcut consistently reduced AUC by 10-15% from baseline across all learning rates, compared to the complex object shortcut (3-7%). This suggests that simpler shortcuts are typically more harmful to model performance, in line with the "simplicity bias" of neural networks (Shah et al., 2020). We also found that reducing the learning rate improved AUC by approximately 8% when the learning rate decreased from 0.01 to 0.001 across all shortcuts. However, even with reduced learning rates, shortcuts in the training data still result in a drop in AUC of 3-11% in the best case (Table 2).

## 5. Discussion & conclusion

We have demonstrated a novel methodology to localize shortcuts to specific model layers using PD distributions and per-layer KL divergence analysis. The approach identifies both

Table 2: ResNet-34 on CheXpert pneumothorax detection task. Table highlights: AUC; $\mathrm{AUC_{diff}} = \mathrm{AUC_{clean}} - \mathrm{AUC}$; total KL divergence from clean model; and mean PD of test set samples (mean $\pm$ std). Performance is assessed across three learning rates: (a) 0.01; (b) 0.001; (c) 0.0001. All models trained with an SGD optimizer.

(a)

| Shortcut | LR | AUC | AUC$_{\mathrm{diff}}$ | Divergence | Mean PD |
|---|---|---|---|---|---|
| Square (constant) | 0.01 | 0.551±0.011 | 0.146±0.045 | 2.728±0.187 | 7.334±0.145 |
| Square (random) | 0.01 | 0.579±0.018 | 0.116±0.019 | 1.179±0.235 | 14.076±0.816 |
| Complex object | 0.01 | 0.621±0.034 | 0.069±0.060 | 0.142±0.040 | 22.755±0.159 |

(b)

| Shortcut | LR | AUC | AUC$_{\mathrm{diff}}$ | Divergence | Mean PD |
|---|---|---|---|---|---|
| Square (constant) | 0.001 | 0.627±0.016 | 0.110±0.011 | 0.538±0.059 | 19.797±0.230 |
| Square (random) | 0.001 | 0.665±0.020 | 0.070±0.014 | 0.278±0.032 | 23.949±0.046 |
| Complex object | 0.001 | 0.701±0.008 | 0.027±0.024 | 0.102±0.006 | 23.550±0.415 |

(c)

| Shortcut | LR | AUC | AUC$_{\mathrm{diff}}$ | Divergence | Mean PD |
|---|---|---|---|---|---|
| Square (constant) | 0.0001 | 0.609±0.018 | 0.132±0.009 | 0.787±0.058 | 19.037±1.247 |
| Square (random) | 0.0001 | 0.645±0.027 | 0.092±0.016 | 0.369±0.113 | 24.500±0.261 |
| Complex object | 0.0001 | 0.688±0.014 | 0.048±0.005 | 0.092±0.018 | 23.796±0.518 |

the samples classified using shortcuts and the network layer responsible for the shortcut classification. The results are consistent across models, datasets, and four synthetic shortcuts. With this, we empirically demonstrate the relationship between visual complexity of shortcuts, the depth of their manifestation in neural networks, and their potential for harm. We find more complex shortcuts manifest deeper in the network than simpler shortcuts, but that simpler shortcuts have greater potential for harm. Careful choice of training conditions can help to reduce the influence of shortcuts, but changes to the optimizer type or learning rate were never enough to completely mitigate the shortcuts in our experiments.

Our investigations rely on access to both clean and shortcut-influenced data. Though not always practical, this approach is useful in situations where a benchmark model is available, such as detecting site-specific shortcuts in multi-site data or testing new training data for shortcuts when updating models. We leave removing this constraint of a clean baseline to future work. Looking ahead, we aim to explore how to build on this work to not just identify shortcuts, but to mitigate their influence. To gain further insight into the clinical applicability of this work, we need to investigate real-world shortcuts using highly curated medical image datasets. This is reserved for future work, but our findings here represent progress towards rigorous evaluation of model safety for clinical deployment.

## Acknowledgments

This work was supported by the UKRI.

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

## Appendix A. Trends across network architectures

In the main body of the paper, we demonstrate that the inclusion of a shortcut in training data changes a network's test set PD distribution. Here, we provide further evidence of our findings through the use of figures that demonstrate the consistency of our results across different network architectures, classification tasks, and optimizer types.

### A.1. SGD optimizer

In Figures 6−8, we showcase the test set PD distributions of our four network architectures trained on each of our clean and shortcut datasets for each classification task using the SGD optimizer. Across each classification task we see a commonality in the shape of the PD distributions for each network architecture trained on the same shortcut dataset.

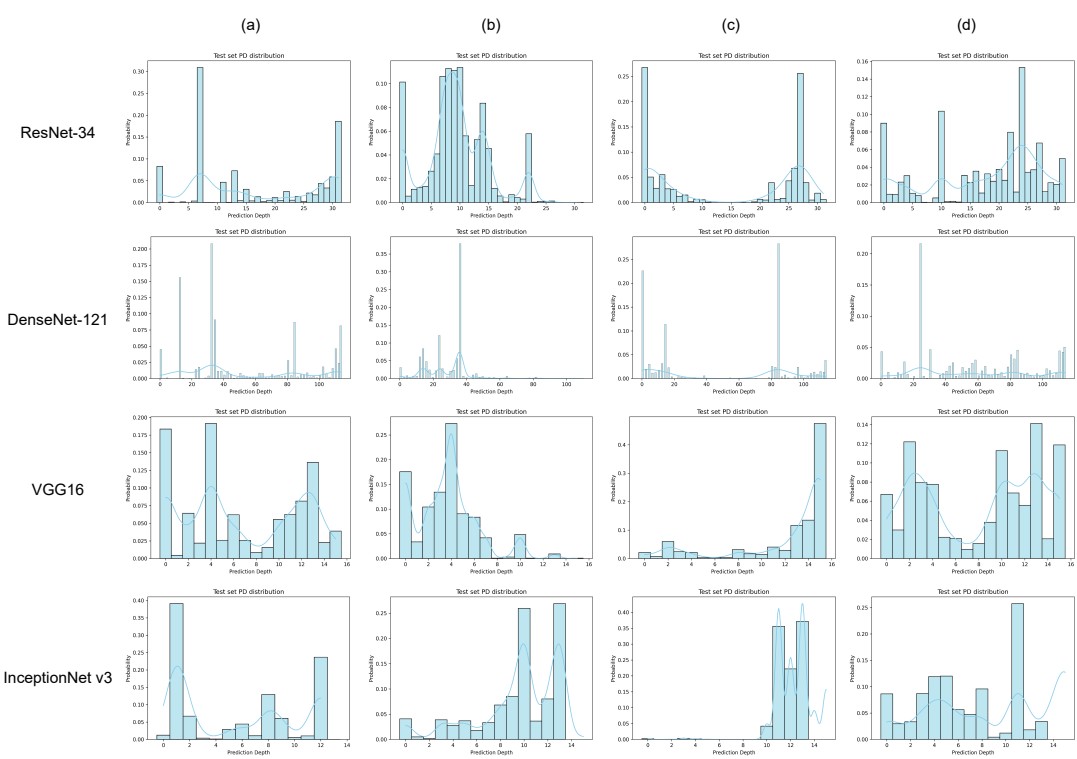

Figure 6: Comparison test set PD distributions of models trained on different versions of the Waterbirds dataset: (a) "clean" benchmark; (b) red square (constant); (c) red square (random); (d) complex object. All networks are trained with an SGD optimizer with a learning rate of 0.01.

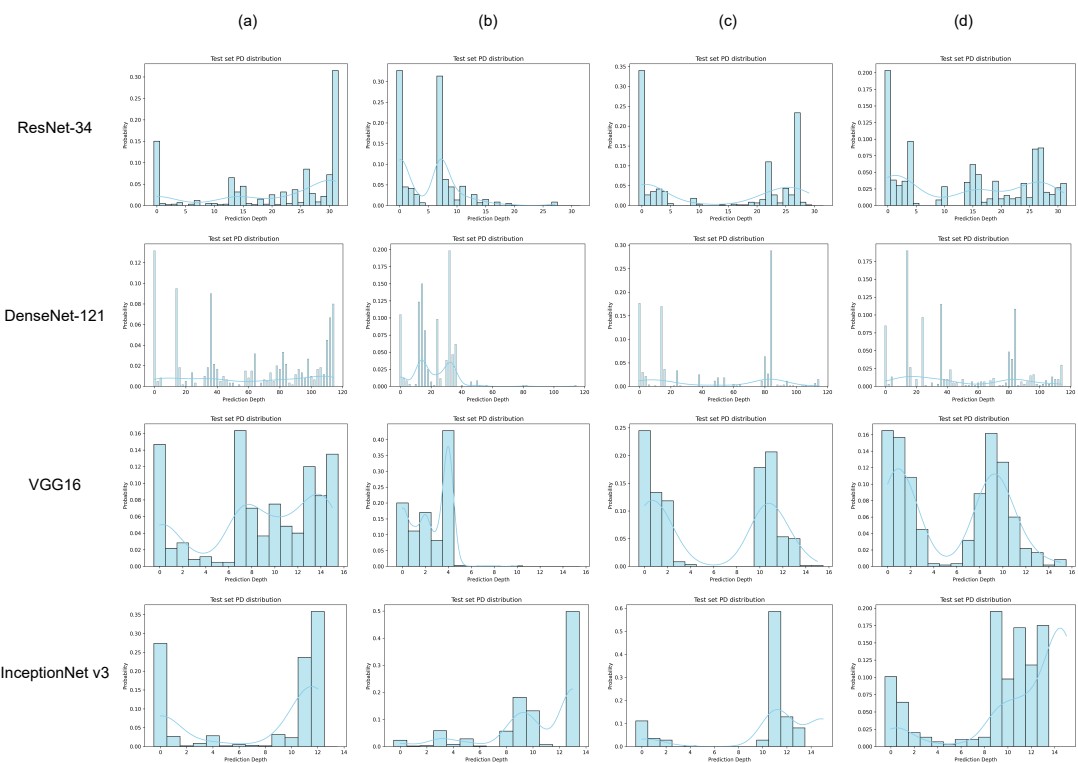

Figure 7: Comparison test set PD distributions of models trained on different versions of the ISIC dataset: (a) "clean" benchmark; (b) red square (constant); (c) red square (random); (d) complex object. All networks are trained with an SGD optimizer with a learning rate of 0.01.

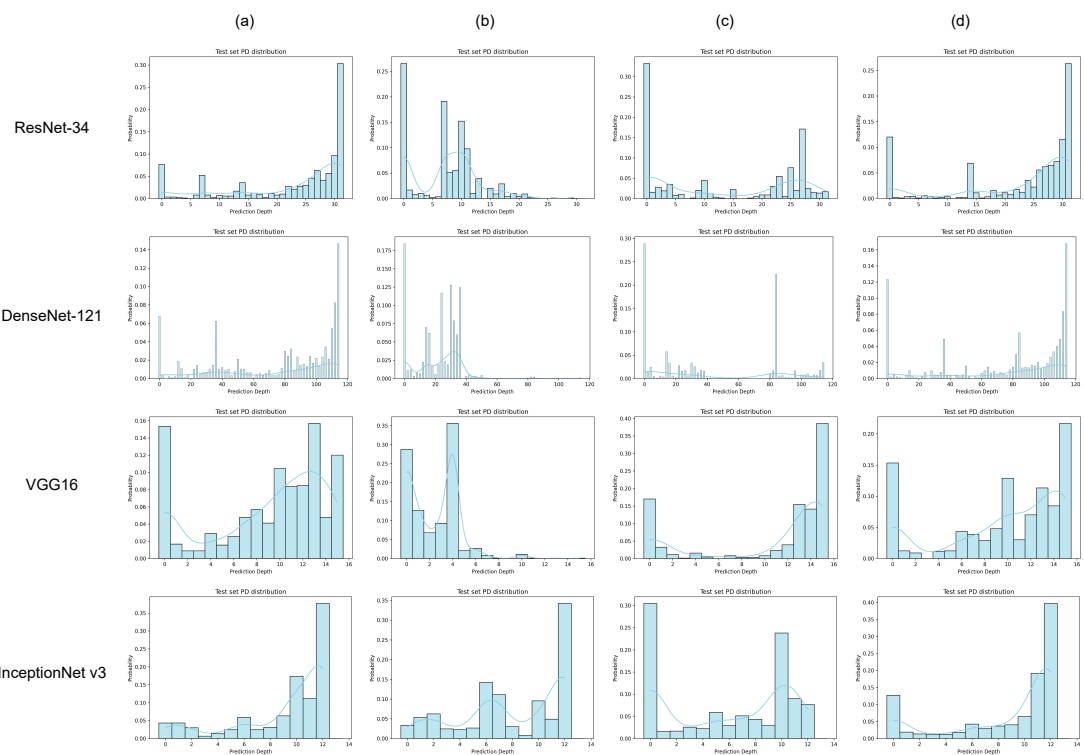

Figure 8: Comparison test set PD distributions of models trained on different versions of the CheXpert dataset: (a) "clean" benchmark; (b) red square (constant); (c) red square (random); (d) complex object. All networks are trained with an SGD optimizer with a learning rate of 0.01.

## A.2. AdamW optimiser

Figures 9−11 show the test set PD distributions for models trained with the AdamW optimizer. Like the models trained with the SGD optimizer, we observe some similarities in the effect of the shortcut across different network architectures. However, with the AdamW optimizer, the impact of the shortcut varies more between architectures and tasks than it did with SGD. Moreover, we notice that some shortcuts have a lesser influence at a higher learning rate (0.01) with the AdamW optimizer, compared to SGD. The differences in the impact of shortcuts between models trained on AdamW and SGD are interesting and motivate further investigation in future work. However, we do not conduct a significant investigation of this here.

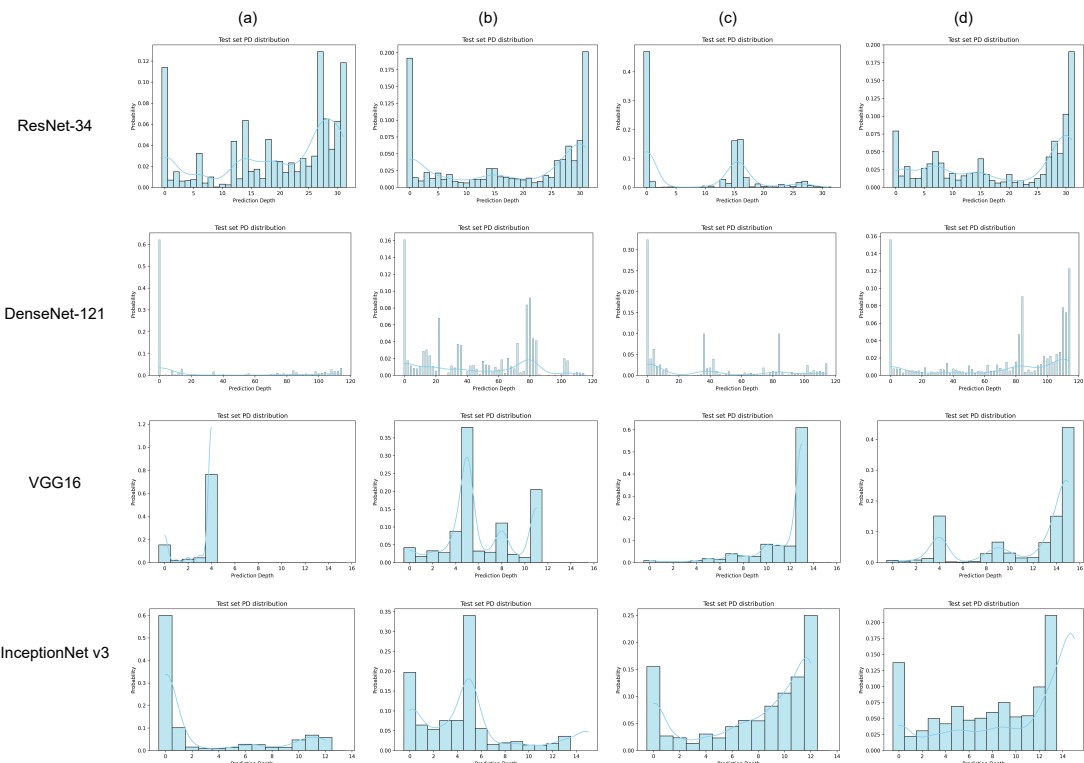

Figure 9: Comparison test set PD distributions of models trained on different versions of the Waterbirds dataset: (a) "clean" benchmark; (b) red square (constant); (c) red square (random); (d) complex object. All networks are trained with an AdamW optimizer with a learning rate of 0.01.

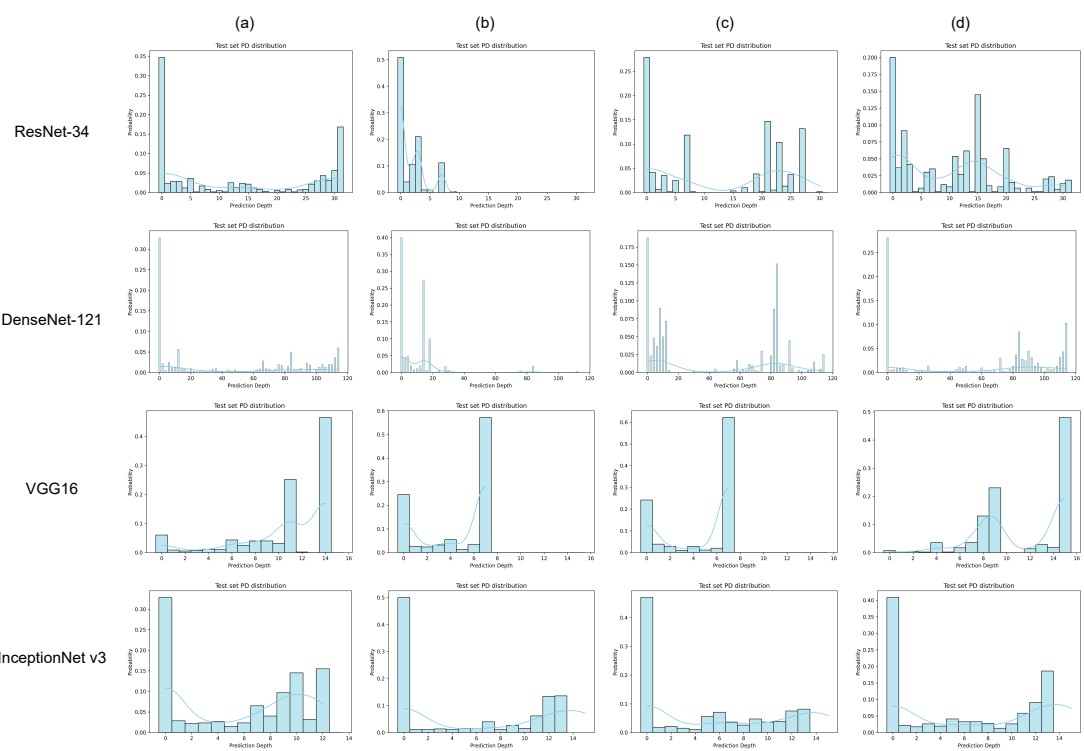

Figure 10: Comparison test set PD distributions of models trained on different versions of the ISIC dataset: (a) "clean" benchmark; (b) red square (constant); (c) red square (random); (d) complex object. All networks are trained with an AdamW optimizer with a learning rate of 0.01.

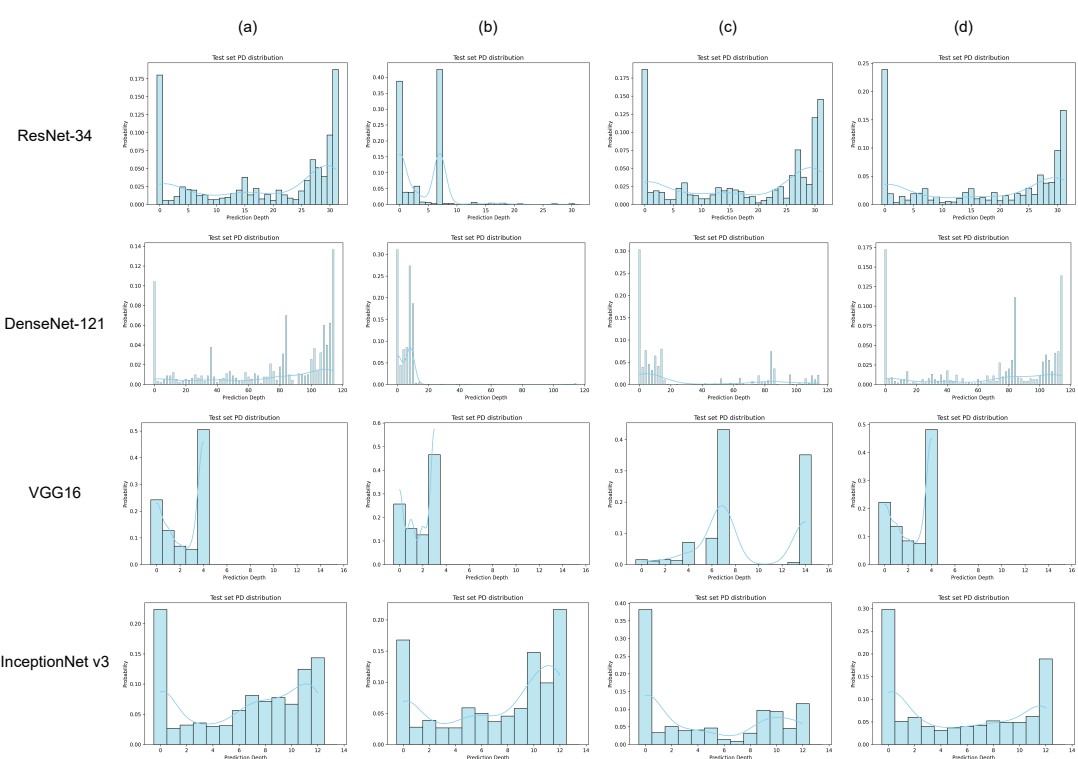

Figure 11: Comparison test set PD distributions of models trained on different versions of the CheXpert dataset: (a) "clean" benchmark; (b) red square (constant); (c) red square (random); (d) complex object. All networks are trained with an AdamW optimizer with a learning rate of 0.01.

## Appendix B. PD of shortcut and no-shortcut samples

In the main body of the paper, we show that shortcut features can be localized to particular model layers. By inspecting samples classified at these layers using saliency maps, we show that shortcut features are used by the models. We expand upon these results by showing the PD distribution of samples with and without the shortcut separately. In many instances, we can observe a clear distinction between the two types of samples, with distinctive peaks in the distribution that are uniquely associated with either shortcut or non-shortcut samples.

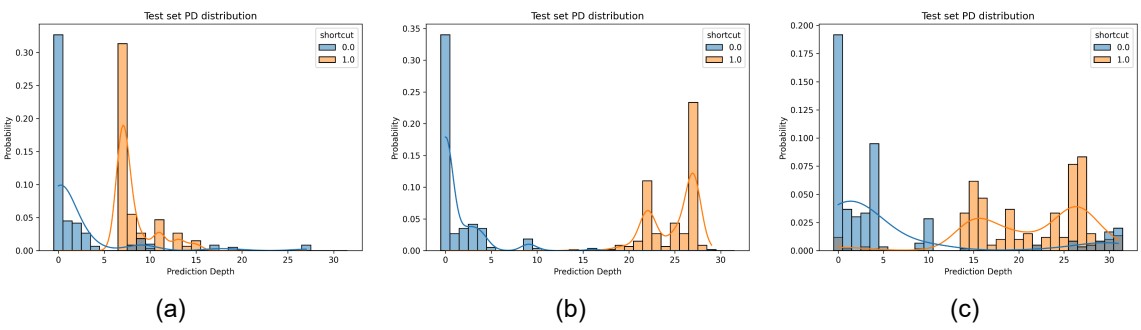

Figure 12: PD distribution of a ResNet34 trained on each of the ISIC datasets featuring synthetic shortcuts: (a) red square (constant location) shortcut; (b) red square (random location) shortcut; (c) complex object shortcut.

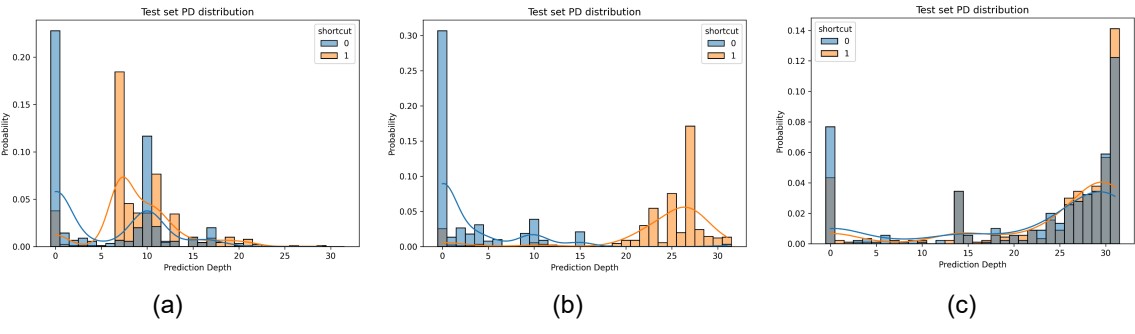

Figure 13: PD distribution of a ResNet34 trained on each of the CheXpert datasets featuring synthetic shortcuts: (a) red square (constant location) shortcut; (b) red square (random location) shortcut; (c) complex object shortcut.

### B.1. Correlation between PD and performance

Here we will elaborate on the correlation between test set PD and model performance, and the correlation between divergence from clean model and performance. Figure 14 displays the correlation between all these values for a DenseNet-121 trained on the CheXpert shortcut datasets and Figure 15 shows the same analysis for a ResNet34 model. We have separated the analysis by the learning rate used during training. We observe a positive correlation between mean test set PD and AUC, and a negative correlation between AUC and divergence from the clean baseline across all learning rates for each model. This suggests that models with deeper PD generally outperform those with shallower PD. This supports our results in Table 2, where we see that the perceptually simpler shortcuts, which cause the most significant decline in mean test set PD, also cause the largest drop in performance from the clean baseline.

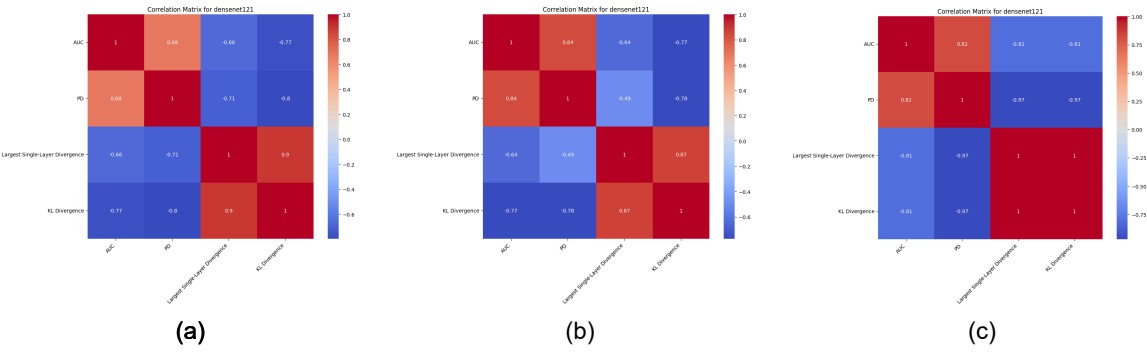

(a)                (b)                (c)

Figure 14: Correlation matrix of the relationship between PD, performance (AUC), and divergence from a baseline model. The network architecture is a DenseNet-121 trained on the ISIC dataset across three learning rates: (a) 0.01; (b) 0.001; (c) 0.0001.

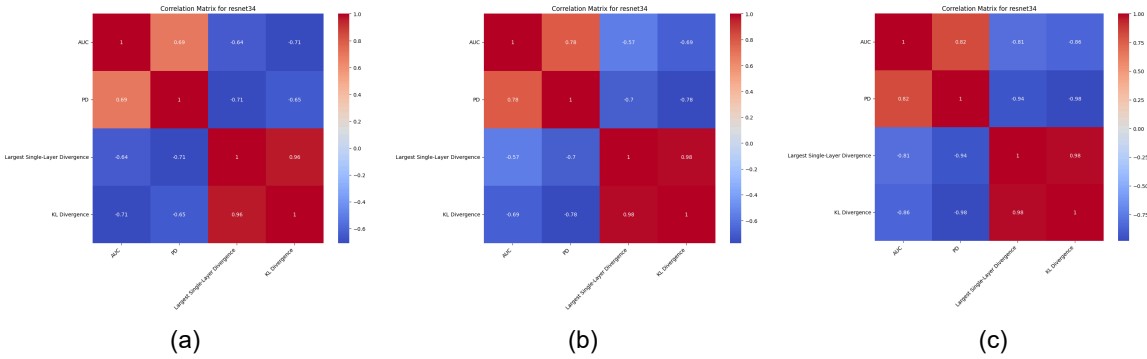

Figure 15: Correlation matrix of the relationship between PD, performance (AUC), and divergence from a baseline model. The network architecture is a ResNet-34 trained on the ISIC dataset across three learning rates: (a) 0.01; (b) 0.001; (c) 0.0001.

