# OpenReview forum: "There Are No Shortcuts to Anywhere Worth Going: Identifying Shortcuts in Deep Learning Models for Medical Image Analysis"
_MIDL.io/2024/Conference — MIDL 2024 Poster_

### Official Review · Reviewer_ZqKW · 2024-02-26

**Confidence:** 4
**Preliminary Rating:** 5
**Recommendation:** Oral
**Final Rating:** 5

**Summary:**

The paper proposes to investigate the feasibility of localizing shortcuts in image classification models. Several datasets are artificially created by adding different types of shortcuts to them. Models are then trained with clean datasets and with datasets containing shortcuts. Prediction depth (PD) is evaluated for each sample of a test set in which the shortcuts are added in balanced way between classes. KL divergence is further used to assess the different PD distributions (with and without shortcuts), to identify the model layers the most impacted by the shortcuts. Finally, the authors show Grad-Cam saliency maps for these layers demonstrating the use of the shortcut in the prediction.

**Strengths:**

The paper is well written. The figures and tables are clear and makes the paper easy to read. The experiments are replicated on several imaging datasets, model architectures and shortcuts type, providing robust results.

**Weaknesses:**

The authors focused on identifying the location of the shortcut within the model but did not comment much about which samples from the test set are classified correctly or not/using the shortcut or not. This would be a very interesting complementary analysis to add or at least to mention in the discussion (see “Questions to address in the rebuttal”).

**Detailed Comments:**

Figure 1 is only referenced after Figure 4 and should appear later in the paper

**Justification Of Final Rating:**

The authors carefully answered my comments and added some analyses to strengthen the results, which allowed me to confirm my preliminary rating. The paper is of good quality and tackles a very interesting topic.

**Justification Of The Preliminary Rating:**

The paper tackles a very interesting problem that is commonly seen in deep learning models on medical data. The approach is innovative, well explained, and tested in many different cases. The results are interesting and properly discussed.

**Questions To Address In The Rebuttal:**

I am wondering what is the difference between the PD distribution of test samples with shortcuts and without shortcuts for each class when classified by the shortcut model. For example in the sentence “Further inspection of samples with a PD of 27 revealed that the model was indeed leveraging the shortcut in order to make its predictions.” Do the authors mean that the samples with a PD of 27 were the ones with the shortcut in the test set?

Is there any correlation between the prediction depth of a sample and if it is classified correctly or not?

**Special Issue:**

Yes

---

> ### Author Response · Authors · 2024-03-15
> **Expanding on PD of shortcut vs no shortcut samples, and correlation between PD and performance**
>
> We thank the reviewer for the time taken to review the paper and provide feedback. We are pleased that they found our approach to be novel and our experiment design and analysis to be robust and well-explained. We appreciate the feedback the reviewer provides to improve and expand upon our analysis to improve the robustness of the paper. All changes to the text are highlighted in red.
>
> > **Figure 1 is only referenced after Figure 4 and should appear later in the paper**
>
> References to figures have been adjusted so that the order of the figures makes logical sense.
>
> > **I am wondering what is the difference between the PD distribution of test samples with shortcuts and without shortcuts for each class when classified by the shortcut model. [...]**
>
> We thank the reviewer for highlighting this analysis, which is currently missing from our paper. We have now included in the appendix Figures 12 and 13, which shows the PD distribution for each of the shortcut models trained on ISIC and CheXpert. We distinguish between the samples that feature a shortcut and the samples that do not. This figure helps to visualise the peaks specific to the shortcut features.
>
> > **Is there any correlation between the prediction depth of a sample and if it is classified correctly or not?**
>
> We find that PD positively correlates with performance metrics (AUC and accuracy). Our experiments find that models with deeper average test set PDs perform better than those with shallow test set PDs. We have added Figures 14 and 15 to the appendix that show the correlation matrices for two different models at different learning rates on the CheXpert dataset. We show that we tend to see a positive correlation between PD and AUC and a negative correlation between PD and divergence from a clean baseline.

---

### Official Review · Reviewer_WM1X · 2024-02-26

**Confidence:** 4
**Preliminary Rating:** 2
**Recommendation:** Poster
**Final Rating:** 3.5

**Summary:**

This paper aims to detect whether shortcut learning is happening in medical image tasks, identify in which specific layers the shortcuts are happening, and propose strategies to mitigate them. Their method relies on prediction depth (PD) and KL divergence to identify changes in the model behaviour after adding a shortcut to the training data. They present results across different type of shortcuts, deep model architectures and datasets. They demonstrate a relationship between the visual complexity of a shortcut, the depth at which its features manifest within the model, and the degree to which the model depends on it.

**Strengths:**

- The use of PD and KL metrics to identify shortcuts is novel.
- Three types of synthetic shortcuts are introduced and evaluated: background, simple and complex object.
- Evaluation on 3 datasets, waterbirds and 2 medical: CheXpert and ISIC.
- Discussion about performance and future work.

**Weaknesses:**

- The introduction and related work provide an insufficient framework for the work presented.
- The choice of metrics (PD and KL) seems arbitrary. The authors provide references and equations but there is no background about why those metrics make sense for this task.
- The use of heatmaps as methods for explainability is not fully reliable. There is a lack of discussion regarding their limitations. Also, the criteria for selecting the heatmaps displayed in figures is not mentioned.
- No code available for transparency and reproducibility of results.

**Detailed Comments:**

- The introduction of Grad-CAM in section 3.1 is missing a reference.

**Justification Of Final Rating:**

Although I believe the introduction and motivation are still rather short, I understand the limitation of pages. I agree that the structure of the manuscript and the readability of the text have been improved.

**Justification Of The Preliminary Rating:**

The experiments and results presented in the paper are interesting. However, the writing could be enhanced to improve the overall presentation, facilitating smoother flow and readability for the audience.

**Questions To Address In The Rebuttal:**

1. The structure and writing of the text could be improved:
    - Introduction is very short, the paper could benefit from further motivation and clearer connections with related works.
    - Some relevant references are missing e.g. Oakden-Rayner, Luke, et al. "Hidden stratification causes clinically meaningful failures in machine learning for medical imaging." *Proceedings of the ACM conference on health, inference, and learning*. 2020.
    - Some important information about the methodology is missing in the abstract: Prediction Depth (PD) and KL divergence metrics.
    - Details about experimental setting appear in “Section 3 Method” and “Section 4: Experiments”. This can be confusing for the readers. I would suggest improving the structure of the sections. A suggestion could be “Section 3: Methods”, “Section 4: Experimental Setting”, “Section 5: Results”.

2. Choice of prediction depth and KL divergence. Further elaborate why these metrics are chosen for your methodology. Any related work worth mentioning?

3. Explainability methods are briefly mentioned in Section 3.1. Justify your choice of heatmaps (GradCAM), and discuss the limitations of saliency maps, see for example Zhang, Jiajin, et al. "Overlooked trustworthiness of saliency maps." International Conference on Medical Image Computing and Computer-Assisted Intervention. Cham: Springer Nature Switzerland, 2022.

**Special Issue:**

No

---

> ### Author Response · Authors · 2024-03-15
> **Updates to the structure and writing, justifying choice of PD and KL divergence, and discussing limitations of explainability methods**
>
> We are very grateful that the reviewer has taken the time to read and provide feedback on our paper. We are pleased that the reviewer finds our approach novel and appreciates the rigour of the experiment design. We thank the reviewer for the suggestions that they have provided to improve the paper’s structure, readability, and robustness. All changes to the text are highlighted in red.
>
> >  1. **The structure and writing of the text could be improved:**
> >      - **Introduction is very short, the paper could benefit from further motivation and clearer connections with related works.**
>
> We have added detail to our introduction (Section 1) to connect our paper to other work investigating shortcut learning. In particular, we highlight two papers that discuss the concept that shortcuts that are harmful to the model are “easy” features with a correlation in the training data, and can be learned in the earliest layers of a network. This connects to our investigation of where, within a neural network, shortcuts of different visual complexities manifest.
>
> > 1. **[...]**
> >    - **Some relevant references are missing e.g. Oakden-Rayner, Luke, et al. "Hidden stratification causes clinically meaningful failures
> in machine learning for medical imaging." Proceedings of the ACM conference on health, inference, and learning. 2020.**
>
> We thank the reviewer for highlighting references relevant to our paper. Oakden-Rayner, Luke, et al. "Hidden stratification causes clinically meaningful failures in machine learning for medical imaging." Proceedings of the ACM conference on health, inference, and learning. 2020, highlights the important issue of hidden stratification. This is particularly relevant to shortcut learning, where the dataset does not label shortcut features. Models can still learn to rely on them, but their identification and mitigation become more challenging. This is an important piece of work to discuss, and we have included it in Section 1.
>
> > 1. **[...]**
> >    - **Some important information about the methodology is missing in the abstract: Prediction Depth (PD) and KL divergence metrics.**
>
> We agree that this information should be included in the abstract and have added it.
>
> > 1. **[...]**
> >    - **Details about experimental setting appear in “Section 3 Method” and “Section 4: Experiments”. This can be confusing for the readers. I would suggest improving the structure of the sections. A suggestion could be “Section 3: Methods”, “Section 4: Experimental Setting”, “Section 5: Results”.**
>
> We thank the reviewers for highlighting weaknesses in the structure of the paper’s writing. We have identified where there is overlap or repeated explanations regarding the experimental setup in Sections 3 and 4. Details that are consistent across all experiments have been moved to Section 3, and the additional experimental detail in each subsection of Section 4 contains information specific to that experiment. This information is discussed immediately before the results to provide a clear flow for readers looking at specific sections.
>
> > 2. **Choice of prediction depth and KL divergence. Further elaborate why these metrics are chosen for your methodology. Any related work worth mentioning?**
>
> In Section 3.1, we motivate our choice for KL divergence by explaining that its asymmetric nature emphasises the layers of the model most influenced by the shortcut’s introduction to the training data. This is beneficial over other metrics such as squared difference.
> The choice of Prediction Depth is motivated by its use as a measure of sample difficulty, connecting to the simplicity bias of deep learning models and shortcuts simplicity over task-specific/clinically relevant features. We added text to Section 2.1 to clarify this connection and connect to literature that connects shortcut learning and PD.
>
> > 3. **Explainability methods are briefly mentioned in Section 3.1. Justify your choice of heatmaps (GradCAM), and discuss the limitations of saliency maps, see for example Zhang, Jiajin, et al. "Overlooked trustworthiness of saliency maps." International Conference on Medical Image Computing and Computer-Assisted Intervention. Cham: Springer Nature Switzerland, 2022.**
>
> We agree that saliency maps have limitations. In Section 3.1, we have included a short discussion of their limitations and literature supporting the use of GradCAM to understand the visual features considered important to a model’s prediction.

---

> > ### Comment · Reviewer_WM1X · 2024-03-26
> >
> > Thanks to the authors for incorporating the feedback. Although I believe the introduction and motivation are still rather short, I understand the limitation of pages. I agree that the structure of the manuscript and the readability of the text have been improved. I have no further questions.

---

### Official Review · Reviewer_N54m · 2024-02-29

**Confidence:** 4
**Preliminary Rating:** 5
**Recommendation:** Oral
**Final Rating:** 5

**Summary:**

The authors present an analysis of shortcut learning in image classification. Their study includes experiments on bird, chest x-ray pneumothorax, and skin lesion classification. The experiments were conducted with three different neural network architectures, three learning rates, and two optimizers. Shortcuts were either naturally present (e.g., background in water bird classification or drains on chest x-rays) or added in a controlled setup.
The study shows that through PD and KL-divergence analysis, one can locate the layers most prone to a given shortcut, having important implications for mitigation strategies. The authors demonstrate that more complex shortcuts can be located deeper inside the network than simple confounders (i.e., red squares in fixed locations).

**Strengths:**

- The paper is written concisely, guiding the reader through the text.
- I appreciate the clear experimental design, thoughtful selection of shortcuts with different complexities, and precise analysis.
- The figures are prepared well and contribute to understanding the method and support the findings described in the text.
- The effort the authors put into benchmarking with three datasets, networks, learning rates, and two optimizers is commendable
- The study contributes to a pressing issue in medical AI.

**Weaknesses:**

- No significant weaknesses in my opinion.
- Since you tested many different setups, I would appreciate it if you could elaborate when describing the findings if you observed common patterns for all tested networks and optimizers (e.g., on the PD vs. shortcut complexity). I am aware of the length constraint; consider, e.g., adding a table showing the results for all experiments or plots for the network architectures you do not showcase in the main text as a supplementary.

**Detailed Comments:**

I appreciate the very thorough experiments and the clear experimental setup. You are able to show the connection between the shortcut complexity and PD nicely.
Since you did many more tests than you can present in the tables/text, it would be good if you could elaborate if and how your findings applied to the tested neural networks, optimizers, etc. (you already mention the influence of the LR on mitigating shortcuts).
I the abstract, you mention that you explore shortcut mitigation strategies. I agree that your work has important implications for mitigation strategies, but do not see the exploration part in the text.

**Justification Of Final Rating:**

I am fully satisfied with the replies and edits by the authors and recommend accepting this contribution. The revision has further improved the text and readability and how has added relevant references.

**Justification Of The Preliminary Rating:**

The study is motivated by a pressing issue in medical AI and puts the effort into the context of prior research. The study is carefully designed and analyzed. The authors conducted experiments on various architectures, learning strategies, and datasets and included shortcuts with different complexities, making the study very relevant for shortcut/confounder mitigation techniques.

**Questions To Address In The Rebuttal:**

- Were the findings the same for the two tested optimizers and for the three tested neural networks?

**Special Issue:**

Yes

---

> ### Author Response · Authors · 2024-03-15
> **Commonality of findings across networks and optimizers, and addressing reference to mitigation in the introduction**
>
> We would like to thank the reviewer for their positive feedback regarding our paper and their helpful suggestions for improvement. We are delighted that the reviewer believes we make a valuable contribution to a pressing issue in medical AI, and that they are pleased with the experimental design and analysis. We appreciate their suggestion to expand upon the discussion of the results to improve the paper. All changes to the text are highlighted in red.
>
>
> > **Were the findings the same for the two tested optimizers and for the three tested neural networks?**
>
> Our results are generally consistent across neural network architectures, and our approach's efficacy to localize learned shortcuts is consistent in each case. We have added to the Appendix (Figures 6-11) visualisation of the PD distributions for each dataset across different architectures, learning rates, and optimiser types. We find that each shortcut appears to have a similar influence on each model. We do notice differences in how the shortcuts manifest across the two optimizer types. Some shortcuts appear to have a lesser influence on PD at high learning rates when training with AdamW, while the opposite is true for SGD. While we don’t perform an extensive investigation of this in this paper, it is an interesting insight to motivate future work.
>
> >  **I the abstract, you mention that you explore shortcut mitigation strategies. I agree that your work has important implications for mitigation strategies, but do not see the exploration part in the text.**
>
> We agree that this sentence misrepresents the insight gained from our work. Here, we were referring to our experiments with different optimizer types and learning rates and how their selection can affect the harm caused by shortcuts. However, we understand that referring to this as "mitigation" is misleading, and we apologize for any confusion this may have caused. We have removed this sentence from the end of the abstract.

---

> > ### Comment · Reviewer_N54m · 2024-03-22
> >
> > Thank you for your responses, I am satisfied with the replies and have no further comments.

---

### Comment · Area_Chair_pv6h · 2024-03-15
**Gentle reminder of rebuttal deadline**

Dear authors,

This is just a gentle reminder that the rebuttal deadline is coming up. I encourage you to take this opportunity so you can engage with the reviewers in the upcoming discussion phase.

Best wishes,
AC

---

### Author Response · Authors · 2024-03-15
**Thank you to the reviewers**

We are grateful to all of the reviewers for their very helpful feedback. We are highly encouraged that they find our proposal to be a novel approach to gaining insight into an important problem in medical image analysis tasks and that each of the reviewers found our experiment design to be thorough and well-evaluated. Answers to every reviewer will be given separately, with references to the modifications made in the paper when appropriate. To note, we made a slight modification to the original ordering of the authors.

---

### Meta-Review · Area_Chair_pv6h · 2024-04-02

**Recommendation:** Accept (Poster)
**Confidence:** 5

**Metareview:**

This paper received mixed reviews with two very positive reviewers. Ultimately all reviewers recommended acceptance, as the more critical reviewer improved their initial rating to borderline accept. I had a closer look and also believe that the authors addressed the raised concerns adequately.

I recommend acceptance of the paper and think it will raise interest in the MIDL community.

However, I would like to point out some template violations (e.g. figure captions, side-by-side figure and table), and urge the authors to adhere to the template rules in the future.

---

### Decision · Program_Chairs · 2024-04-06

Accept (Poster)